# Optimized Lambda Exonuclease Digestion or Purification Using Streptavidin-Coated Beads: Which One Is Best for Successful DNA Aptamer Selection?

**DOI:** 10.3390/mps5060089

**Published:** 2022-10-29

**Authors:** Lisa Lucie Le Dortz, Clotilde Rouxel, Quentin Leroy, Noah Brosseau, Henri-Jean Boulouis, Nadia Haddad, Anne-Claire Lagrée, Pierre Lucien Deshuillers

**Affiliations:** Laboratoire de Santé Animale, Anses, INRAe, Ecole nationale vétérinaire d’Alfort, UMR BIPAR, F-94700 Maisons-Alfort, France

**Keywords:** aptamer, ssDNA generation, SELEX, streptavidin-coated beads, lambda exonuclease digestion

## Abstract

The high failure rate of the in vitro aptamer selection process by SELEX (Systematic Evolution of Ligands by EXponential enrichment) limits the production of these innovative oligonucleotides and, consequently, limits their potential applications. The generation of single-stranded DNA (ssDNA) is a critical step of SELEX, directly affecting the enrichment and the selection of potential binding sequences. The main goal of this study was to confirm the best method for generating ssDNA by comparing the purification of ssDNA, using streptavidin-coated beads, and lambda exonuclease digestion, and by improving ssDNA recovery through protocol improvements. In addition, three techniques for quantifying the ssDNA generated (Qubit vs. Nanodrop^TM^ vs. gel quantification) were compared, and these demonstrated the accuracy of the gel-based quantification method. Lambda exonuclease digestion was found to be more efficient for ssDNA recovery than purification using streptavidin-coated beads, both quantitatively and qualitatively. In conclusion, this work provides a detailed and rigorous protocol for generating ssDNA, improving the chances of a successful aptamer selection process.

## 1. Introduction

In the 20th century, nucleic acids were mainly studied for their role as carriers of genetic information. In 1990, Ellington and Szostak published a pioneering article related to the binding specificity of RNA to ligands and to a method for their synthesis and isolation [1]. They proposed the term “aptamers” to designate them. In the same year, Tuerk and Gold described an in vitro selection method for the production of aptamers with high affinity, which they called “SELEX” for Systematic Evolution of Ligand by EXponential enrichment [2]. SELEX is still the reference method used today. Aptamers are defined as short single-stranded DNA or RNA sequences (generally <100 nucleotides) able to bind to a range of targets with high affinity and specificity, such as proteins [3], lipids, small inorganic molecules, co-factors, cells [4], bacteria [5], and viruses [6], through various interactions, for example, hydrogen bonds, electrostatic and Van der Waals (VdW) interactions [7]. Aptamers are also known as chemical antibodies with significant advantages in terms of stability, synthesis, target diversity and non-toxicity [8]. Functioning as ligands with agonistic or antagonistic properties, they have been developed in many research fields, such as oncology [9], virology, and microbiology, mainly for therapeutic [10,11], diagnostic, and imaging purposes [12,13]. Despite the growing interest in the scientific community for aptamers over the past 25 years, only one aptamer drug was approved by the FDA in 2004 for the treatment of neovascular, age-related, macular degeneration [11]. This limited development can be partly explained by the complex and time-consuming selection process of aptamers. SELEX is an iterative selection process involving several steps, including: incubation of ssDNA/RNA random library with a target, PCR amplification of the binding sequences, and generation of ssDNA/RNA. Despite its apparent simplicity, SELEX has a high failure rate, possibly as high as 70% [14]. The generation of ssDNA is a critical step, requiring a fast and efficient method to produce enough ssDNA that are free of impurities, for the enrichment of potentially binding sequences [15].

Several ssDNA generation methods have been reported for SELEX, including asymmetric PCR, strand separation under denaturing conditions, magnetic separation with streptavidin-coated beads and lambda exonuclease digestion [16,17]. Asymmetric PCR is often considered a robust and low-cost method [16,18]. However, this method is time-consuming as it is strongly recommended to adjust the PCR conditions for each round of SELEX. Several optimizations can be performed, such as the modification of primers concentration and/or ratio, number of PCR cycles, annealing temperature, dNTP, and MgCl_2_ concentrations [19,20]. In addition, asymmetric PCR produces ssDNA, as well as dsDNA [18,21]. As a result, asymmetric PCR is often combined with a gel purification step or other ssDNA generation methods [22]. Strand separation under denaturing conditions is based on the differential migration of two strands of different sizes, and the desired band is then purified from the gel. Although very efficient, this method is time-consuming and requires the use of more expensive primers with chemical modifications [16]. Purification of ssDNA using streptavidin-coated beads and lambda exonuclease digestion are frequently used in the literature as they produce less or no dsDNA, and do not require a gel purification step. However, the removal of streptavidin or lambda exonuclease by phenol-chloroform extraction is advised, potentially significantly decreasing the amount of recovered ssDNA [23,24].

Very few studies have compared the lambda exonuclease digestion and purification of ssDNA using streptavidin-coated beads for ssDNA generation [25,26]. Those comparison studies used standard ssDNA generation protocols, without investigating and applying optimizations proposed by other studies to improve the final ssDNA yield [21,27]. In addition, only one study has reported the yields obtained after digestion by lambda exonuclease and phenol-chloroform extraction, and no yield data are available for streptavidin-coated beads [24]. This lack of information makes the choice of the ssDNA generation method difficult for aptamer selection. To overcome this gap in knowledge, the quality and quantity of the ssDNA generated for purification using streptavidin-coated beads and lambda exonuclease digestion were compared, before and after ethanol precipitation or phenol-chloroform extraction. The main goal of this paper is to provide a rigorous comparison of ssDNA generation methods by addressing the following aspects: protocol optimizations, study of the impact of ethanol precipitation and phenol-chloroform extraction on ssDNA recovery, and the choice of an accurate ssDNA quantification technique, by comparing three techniques: absorbance at 260 nm (Nanodrop™), fluorimetry (Qubit), and image analysis following gel migration (ImageJ).

## 2. Materials and Methods

### 2.1. ssDNA Library and Primers

A ssDNA library was purchased from Eurogentec, after polyacrylamide gel electrophoresis purification (PAGE). This library was made up of approximately 6 × 10^14^ different ssDNA sequences, consisting of 40 random nucleotides flanked by 20 nucleotide-long constant sequences for primer hybridization (5′-CTCCTCTGACTGTAACCACG40NGCATAGGTAGTCCAGAAGCC-3′ [28]). The library was amplified by PCR using a common forward primer (5′-CTCCTCTGACTGTAACCACG-3′; Eurofins). A biotinylated reverse primer (5′-Biotin GGCTTCTGG ACTACCTATGC-3′; Eurofins), purified by HPLC, was used for purification using streptavidin-coated beads, while a phosphorylated reverse primer (5′-Phosphate GGCTTCTGGACTACCTATGC-3′; Eurogentec), purified by HPLC with unprotected phosphate groups, was needed for lambda exonuclease digestion. The concentration of the primers and the library was determined by the supplier, by measuring the absorbance at 260 nm and calculated from the molar extinction coefficient of the DNA. Oligonucleotides were stored at a concentration of 100 µM at −20 °C.

### 2.2. Amplification of ssDNA Library

#### 2.2.1. Optimization of PCR Conditions

An annealing temperature gradient (from 50 °C to 70 °C) and a number of amplification cycles (6, 8, 10, 12, 15, 20 and 25 cycles) were tested independently to define the optimal PCR conditions. The ssDNA library was denatured at 95 °C for 5 min and 2.5 µL of 1 µM library (10^12^ sequences) was used as a starting template. For each amplification, PCR reactions were conducted in a volume of 25 µL, including GoTaq Flexi Buffer 1× (Promega, Madison, WI, USA), 200 µM of dNTPs (Invitrogen, Waltham, MA, USA), 2.5 mM of MgCl_2_ (Promega), 1.25 units of GoTaq G2 Hot Start Polymerase (Promega), and 0.5 µM of forward and modified reverse primers. The amplification program consisted of an initial DNA denaturation at 95 °C for 2 min, followed by 6, 8, 10, 12, 15, 20 and 25 cycles of denaturation at 95 °C for 30 s, annealing at 50 °C to 70 °C for 30 s and elongation at 72 °C for 15 s. A final extension was performed at 72 °C for 60 s. PCR products (10 µL) with 2 µL of loading buffer (Invitrogen) were run on a 4% (*w*/*v*) agarose gel stained with GelRed^®^ (Biotium, Fremont, CA, USA) in TAE 1× buffer (Bioscience, Allentown, PA, USA) at 90 V for 2 h. A 50 bp size marker (TrackIt™50 bp DNA Ladder, Invitrogen) was used for size determination.

#### 2.2.2. Large-Scale PCR Amplification

The optimal reaction conditions were applied to perform large-scale PCR amplification with 40 PCR tubes, each containing 50 µL (for a final volume of 2 mL). PCR products were purified using a NucleoSpin Gel and a PCR Clean Up Kit (Macherey-Nagel, Düren, Germany), according to the manufacturer’s instructions, with the following modifications: elution was performed twice after 5 min of incubation with 15 µL of elution buffer, previously heated to 70 °C. Ten µL of PCR products were diluted to 1/5th and deposited on a 4% (*w*/*v*) agarose gel. The dsDNA concentration was determined after gel electrophoresis with a quantitative size marker (Quick-Load Purple Low Molecular Weight DNA Ladder, NEB) and electrophoretic analysis of the band intensities using the ImageJ program.

### 2.3. Optimization of Capture on Streptavidin-Coated Magnetic Beads

Dynabeads M-270 (Thermo Fisher Scientific, Waltham, MA, USA) were used as previously described [29], with the amount of beads recommended (20 mg of beads for 2 µg of dsDNA). Before their use, the beads were washed three times with a 2× Binding and Washing buffer (10 mM Tris-HCl, 1 mM EDTA and 2 M NaCl). The biotinylated PCR products (2 µg) were incubated with pre-washed beads for 30 min at 850 rpm. To improve recovery of the non-biotinylated sense DNA strand, alkaline and thermal denaturation were compared. For alkaline denaturation, beads were incubated twice with 50 µL of a freshly prepared 200 mM NaOH solution for 3 min. The eluted ssDNA was neutralized with TE buffer (pH = 7.5) and HCl (150 mM). For thermal denaturation, the bead suspension was heated to 95 °C for 15 min at 850 rpm [29]. For both methods, the eluted products were revealed on a 4% (*w*/*v*) agarose gel stained with GelRed^®^ and a 50 bp marker. For further optimization, the number of beads was increased (50 mg for 2 µg of dsDNA) and a range of NaOH concentrations was tested (20, 75, 150 and 300 mM) for alkaline elution.

### 2.4. Optimization of Lambda Exonuclease Digestion

For the lambda exonuclease digestion of the anti-sense DNA strand, digestion time is the most critical step [21]. Hence, several incubation times (5, 15, 30, 45, and 60 min) were tested. For each time point, 2 µg of phosphorylated dsDNA were incubated with 5 μL of reaction buffer (Thermo Fisher Scientific, Waltham, MA, USA), and 1 µL of lambda exonuclease (10 U/µL, Thermo Fisher Scientific, Waltham, MA, USA), filled to a final volume of 50 µL with ultrapure water. The enzymatic mix was incubated at 37 °C at 650 rpm. The enzyme was then heat-inactivated (80 °C for 15 min). The products were revealed on a 4% (*w*/*v*) agarose gel stained with GelRed^®^, using a 50 bp size marker for size determination. The incubation time giving the highest dsDNA digestion was chosen for the method comparison.

### 2.5. Comparison of ssDNA Generation Methods

To compare ssDNA generation with streptavidin-coated magnetic beads or lambda exonuclease digestion, optimal conditions were set as follows: 2 µg of dsDNA template, 50 mg of beads and alkaline denaturation with 150 mM NaOH for the first method, and 30 min of digestion at 37 °C for the second method.

#### 2.5.1. Purification of ssDNA

For each method, ssDNA was purified by ethanol precipitation (n = 5 samples) or phenol-chloroform extraction (n = 5 samples). For ethanol precipitation, a linear polyacrylamide co-precipitant (5 µL, Invitrogen), 1/10th of the sample volume of 3 M sodium acetate (pH = 5.2), supplemented with 2.5 volumes of absolute ethanol at −20 °C, were added to the eluted products and incubated overnight at −20 °C. After centrifugation for 20 min at 20,000× *g* at 4 °C, the pellets were washed with 650 μL of 95% ethanol [30]. Centrifugation was performed for 15 min at 20,000× *g* at 4 °C and the final pellet was dissolved in 50 µL of ultrapure water after drying for 15 min. For phenol-chloroform extraction, one volume of phenol: chloroform: isoamyl alcohol (25:24:1, Sigma Aldrich, Saint-Louis, MO, USA) was added to the eluted products. After centrifugation for 5 min at 16,000× *g* at 4 °C, the aqueous upper phase was recovered for each sample and precipitated with ethanol, following the protocol described above. The final products were revealed on a 4% (*w*/*v*) agarose gel stained with GelRed^®^.

#### 2.5.2. ssDNA Quantification Techniques

The ssDNA generated was quantified by three techniques: fluorimetry with Qubit, absorbance at 260 nm with Nanodrop^TM^, and gel-based quantification with ImageJ software. Two Qubit measurement kits for Qubit Fluorometer 4 (Thermo Fisher Scientific, Waltham, MA, USA) were used, according to the manufacturer’s instructions. The amount of ssDNA in the samples was determined by subtracting the amount of dsDNA, measured by Qubit dsDNA HS (which only detects dsDNA), from the amount of total DNA, measured by Qubit ssDNA (which detects both dsDNA and ssDNA). ssDNA was quantified spectrophotometrically (Nanodrop One, Thermo Fisher Scientific, Waltham, MA, USA) at 260 nm. For gel-based quantification, the final products (10 µL), mixed with 2 µL of loading buffer 6×, were loaded into a 4% (*w*/*v*) agarose gel stained with GelRed^®^, along with the initial library at various concentrations (4, 8, 12, 16 and 20 ng/μL). Electrophoresis was run in TAE 1× buffer for 2 h at 90 V. A calibration curve was designed with the intensity of the library bands plotted against library concentrations. ssDNA concentrations of eluted or digested samples were calculated from band intensities. To validate the concentration range and method, a sample with a known ssDNA concentration from the library (26 ng/µL) was used as a control (Appendix A).

#### 2.5.3. ssDNA Recovery

The method yields were determined using the three ssDNA quantification techniques, before and after the purification step with the following formula: $\%\;\mathrm{ssDNA}\;\mathrm{recovery}=\left(\frac{\mathrm{amount}\;\mathrm{of}\;\mathrm{ssDNA}\;\mathrm{obtained}\;\mathrm{before}\;\mathrm{or}\;\mathrm{after}\;\mathrm{extraction}\;\mathrm{step}}{\mathrm{total}\;\mathrm{amount}\;\mathrm{of}\;\mathrm{ssDNA}\;\mathrm{before}\;\mathrm{purification}:1\;µg}\right)\times 100$. The initial amount of dsDNA used for both methods is 2 µg, corresponding to 1 µg of ssDNA.

#### 2.5.4. Statistical Analysis

All the variables were expressed as mean ± standard deviation (SD). A Student’s *t*-test was performed to compare the ssDNA amounts recovered by the different methods. The results were considered statistically significant for *p* < 0.05.

## 3. Results

### 3.1. Optimization of PCR Conditions

In order to optimize the binding of primers with the library, annealing temperatures ranging from 50 °C to 70 °C were tested (Figure 1). Between 50 °C and 64 °C, a band estimated by ImageJ software at 95 bp was observed. Above 64 °C, a lower band estimated at 87 bp (ImageJ software) became predominant (Figure 1a). The presence of these two bands at unexpected sizes is explained by the important number of cycles (20 cycles) applied for the temperature range. The choice of annealing temperature was based on the yield of dsDNA. An annealing temperature of 58 °C was considered optimal and used further for PCR, as it allowed a maximum amount of dsDNA (Figure 1b) to be obtained. To set the optimal number of amplification cycles, the library was amplified for 6, 8, 10, 12, 15, 20 and 25 cycles at 58 °C (Figure 2). An upper band (95 bp) appeared as early as 8 cycles of amplification and became the most prominent band from the 15th cycle. From the 10th amplification cycle, the gel revealed a band at 40 bp (ssDNA), corresponding to non-amplified sequences due to the limitation of PCR reagents. Six amplification cycles were considered optimal based on the production of dsDNA at the expected size (80 bp). In order to validate these results, a new temperature range was performed from 40 °C to 70 °C with the optimal number of cycles (6 cycles, Figure 3). At low (<51 °C) and high temperature (>61 °C), a decrease of PCR yield was observed. The amplification of 0.1 µM of library at 58 °C after 6 cycles allowed the production of large quantities of dsDNA at the expected size, confirming that these PCR conditions were optimal.

### 3.2. Optimization of Capture on Streptavidin-Coated Magnetic Beads

To improve ssDNA recovery, alkaline and thermal denaturation were compared (Figure 4). With alkaline elution (200 mM NaOH), ssDNA and dsDNA were eluted in the same proportions. With thermal denaturation, dsDNA and high molecular weight structures, corresponding to a complex of streptavidin and biotinylated DNA, were recovered. Contrary to alkaline denaturation, ssDNA was not eluted by thermal denaturation in our study (Figure 4a). In view of these initial results, thermal elution was abandoned, and optimization of alkaline elution was pursued. Specifically, an increased quantity of beads (from 20 mg to 50 mg for 2 µg of DNA) and several NaOH concentrations (from 20 to 300 mM) were tested in order to improve the capture of PCR products and the recovery of ssDNA, respectively. With an excess number of beads relative to dsDNA (50 mg for 2 µg of dsDNA), the capture of the PCR product was very efficient since no band corresponding to the PCR product could be detected after the capture and washing steps (data not shown). For all NaOH concentrations, the predominantly eluted DNA was ssDNA. However, for NaOH concentrations above 200 mM, dsDNA contamination was observed (Figure 4b), indicating the dissociation of the biotin-streptavidin bond. Comparison of the intensity of the bands with ImageJ software showed that 150 mM of NaOH allowed for a better ssDNA recovery compared to 20 or 75 mM NaOH (Figure 4b). This concentration was therefore considered optimal and was used for the following experiments.

### 3.3. Optimization of Lambda Exonuclease Digestion

Lambda exonuclease digestion of the phosphorylated strand was confirmed by a time-course analysis after 5, 15, 30, 45 and 60 min of incubation (Figure 5). After 5 min of incubation with lambda exonuclease, two bands corresponding to ssDNA and dsDNA were revealed, indicating partial digestion. From 15 min, only ssDNA was detected, and the intensity of the bands remained unchanged between 15 and 60 min of digestion (ImageJ). In line with the literature, a 30 min incubation time was selected to ensure an optimal digestion of the phosphorylated strand [21].

### 3.4. Comparison of ssDNA Quantification Techniques

Three different quantification techniques, including fluorimetry (Qubit), absorbance at 260 nm (Nanodrop™), and gel quantification, were compared. Pre-extraction yields cannot be determined by Nanodrop or Qubit as the presence of streptavidin or lambda exonuclease and salts interfere considerably with the assays. For this reason, only the post-extraction yields were compared. Under these conditions, the yields determined by Qubit, Nanodrop^TM^ and gel-based quantification were significantly different according to Student’s *t*-test (*p* < 0.05) (Appendix A, Figure 6 and Figure 7). Regardless of the method used to generate ssDNA, the Nanodrop technique provided the highest values, followed by the gel quantification method, then by the Qubit method. The use of a sample with a known ssDNA concentration (26 ng/µL) demonstrated the accuracy of the gel-based quantification method; therefore, this method was chosen for the comparison of ssDNA generation yields (Appendix A).

### 3.5. Comparison of ssDNA Generation Methods

#### 3.5.1. Quantitative Evaluation

The recovery rates of ssDNA quantified by gel electrophoresis after purification using streptavidin-coated beads or lambda exonuclease digestion were compared (Figure 8, Appendix A). Before ethanol or phenol-chloroform extraction, the recovery rate of ssDNA was estimated at 44% ± 5.8 and 74% ± 9.3, using streptavidin-coated beads and lambda exonuclease digestion, respectively. After ethanol precipitation, the yield of ssDNA decreased significantly to 34.3% ± 6.4 (n = 5) for the streptavidin-coated beads, and to 58.1% ± 3.9 (n = 5) for lambda exonuclease digestion. These yields were not significantly different after phenol-chloroform extraction with 39.7% ± 1.8 (n = 5) for the streptavidin-coated beads, and 56.6% ± 6.0 (n = 5) for lambda exonuclease digestion. Overall, lambda exonuclease digestion, with or without extraction steps, resulted in a 20–30% higher ssDNA recovery than purification using streptavidin-coated beads.

#### 3.5.2. Qualitative Evaluation

In order to purify and concentrate the final product, two purification methods were compared (phenol-chloroform extraction vs. ethanol precipitation) to aid in the choice of the purification method. The results showed no significant difference between the two methods (Student’s *t*-test, *p* > 0.05). Phenol-chloroform extraction did not significantly change the recovery rate compared to simple ethanol precipitation (Figure 8, Appendix A). In addition, it allowed for the elimination of lambda exonuclease and streptavidin, still present after ssDNA generation.

After the purification of ssDNA using streptavidin-coated beads, undesired dsDNA was observed (Figure 4a) and only ssDNA was detected after 30 min of digestion by lambda exonuclease (Figure 5). Lambda exonuclease allowed the generation of ssDNA free of impurities with less dsDNA contamination compared to the streptavidin-coated beads.

## 4. Discussion

A starting SELEX library consists of numerous sequences, composed of variable regions with more or less complexity, making aptamer amplification difficult and problematic. The annealing temperature was chosen based on the maximum dsDNA yield as it is reported to have little influence on the presence of non-specific products [18]. We observed a maximum amount of dsDNA at 58 °C (Figure 1b), which was the theoretical annealing temperature of the primers, as determined in silico. For the following experiments, the annealing temperature was set at 58 °C. The presence of the two bands at unexpected sizes was attributed to the unusually high number of PCR cycles for aptamer amplification (20 cycles). Then, we optimized the number of PCR cycles, as recommended in the literature, to avoid PCR by-products, as they increase with the numbers of PCR cycles [31,32]. Importantly, sequences can hybridize to each other by complementarity, with their variable region or their constant regions forming larger size products, which become more and more prevalent as the number of amplification cycle increases [19]. Starting with a concentration of 0.1 µM of library, in our conditions, the optimal number of amplification cycles corresponded to 6 cycles, as it produces dsDNA at the expected size, which is consistent with the literature (Figure 2) [24]. Using a temperature range from 40 °C to 70 °C, we finally confirmed the optimal annealing temperature for 6 cycles of amplification (Figure 3). We observed a decrease of PCR yields at low and high annealing temperatures. At a low annealing temperature, primers can hybridize non-specifically to the complementary sequences present in the variable regions, reducing the efficiency of the PCR [19]. High annealing temperature weakens the hybridization of primers at their binding sites, also decreasing the PCR efficiency [18]. The annealing temperature can be set to 58 °C for aptamer selection with our library. However, the number of amplification cycles, depending on the number of aptamers selected, needs to be optimized at each selection round.

Before comparing the two ssDNA generation methods, their respective protocols were optimized to improve ssDNA recovery. For streptavidin-coated beads, three improvements were tested. First, the recovery of the non-biotinylated sense strand by alkaline treatments gave better yields and the lowest streptavidin contamination, compared to the heat treatment, which may rupture the non-covalent bonds of streptavidin as previously documented [27] (Figure 4a). Second, an increase of 25 to 50 mg of beads improved the capture of 2 µg of PCR products and downstream ssDNA recovery. Third, dsDNA contamination was observed at a high NaOH concentration (200 mM), due to the breakdown of hydrophobic, VdW or hydrogen bonds or the surface loop between biotinylated DNA and streptavidin (Figure 4b) [33]. As sensitivity of the beads to the alkaline treatment depends on their three-dimensional structure and their steric surface, a range of NaOH concentration needs to be tested for each type of bead [33]. In view of our results, a NaOH concentration of 150 mM is strongly recommended for users of Dynabeads M-270. The most critical step for lambda exonuclease digestion is the incubation time. Based on a time-course analysis, we obtained an optimal digestion at 30 min for 1 µg of phosphorylated strand by 10 unit/µL of lambda exonuclease (Figure 5). Previous studies report digestion of the other strand when incubation time is increased, a phenomenon which was not observed in our case [24]. Lambda exonuclease works in several buffers, but not with the same efficiency [34]. Exploring the efficiency of lambda exonuclease digestion in PCR mix would be worthwhile and may help saving precious time.

After the optimization of ssDNA generation, the amount of ssDNA recovered with each method was compared and quantified using three ssDNA quantification techniques (Figure 6 and Figure 7). Most comparison studies quantify ssDNA recovery by absorbance at 260 nm or by gel electrophoresis. However, to the best of our knowledge, no study has compared the three ssDNA quantification techniques described herein. The gel-based quantification technique has many advantages. This technique allows the visualization of dsDNA and ssDNA, when using GelRed^®^ as an intercaling agent. GelRed^®^ is less efficient to detect ssDNA than dsDNA and the recognition is dependent on the type of sequences [35]. However, GelRed^®^ is much more sensitive than ethidium bromide for both dsDNA and ssDNA detection, as ethidium bromide detects ssDNA very weakly [35]. According to our results, this intercaling agent could be strongly recommended to quantify ssDNA with gel electrophoresis. In addition, the gel-based quantification avoids interference by salt or organic solvents for ssDNA quantification, as only ssDNA band intensity is analyzed by ImageJ software. Qubit is often considered a reliable and repeatable measurement method for dsDNA quantification [36]. However, the specificity and sensitivity of Qubit for ssDNA detection has not yet been demonstrated. In our study, an underestimation of ssDNA concentration was observed compared to the gel quantification method, which can be explained by the presence of phenol/ethanol residues in the sample after precipitation, already observed after Trizol extraction [36]. Based on these results, the Qubit technique appears to be unreliable for the quantification of ssDNA. The overestimation of ssDNA concentrations observed with Nanodrop^TM^ is not surprising, particularly in cases of low concentration samples (<20 ng/µL) and contamination by organic solvents. Ethanol shows UV absorbance with peak values at 230–240 nm, while phenol contamination shows a clear shift of the 260 nm peak in the spectrum towards 270 nm [37]. In addition, the Nanodrop^TM^ cannot distinguish dsDNA from ssDNA, oligonucleotides and free nucleotides. Thus, gel electrophoresis is demonstrated to be the most accurate ssDNA quantification method among the three methods tested. Another accurate method for ssDNA quantification is called Enzyme-linked oligonucleotide assay (ELONA) [25,26]. However, compared to the gel quantification, it is very expensive and time-consuming, and therefore not adequate for routine, rapid quantification.

The yields quantified by gel electrophoresis after streptavidin-coated bead vs. lambda exonuclease generation were compared (Figure 8). The yields of streptavidin-coated beads, quantified before purification (44%), were lower than those observed in the study of Civit et al., (62%) [26]. This may be explained by the use of different magnetic beads (Dynabeads MyOne C1), with a better binding capacity. However, from the available publications, these beads seem less frequently used for SELEX. After ethanol precipitation (34.3%), the ssDNA recovery was lower than in previously published data: Liang et al. obtained a yield over 80%, quantified by Nanodrop, but this method overestimates ssDNA concentrations [38]. Our study provides new data on the yield of streptavidin-coated beads after phenol-chloroform extraction (39.7%), as no published study mentions it. In the case of lambda exonuclease digestion, the ssDNA recovery rate obtained before purification (74%) is higher than the rates reported in the literature (65% in the study of Civit et al.) [26]. These positive results can be attributed to the optimization of PCR conditions and the determination of an optimal lambda digestion time. After phenol-chloroform extraction, the yield (56.6%) was higher than that obtained by Citartan et al., (39.19% by absorbance measurement at 260 nm) [24]. Regardless of the assay technique, lambda exonuclease digestion provided higher yields compared to the purification of ssDNA using streptavidin-coated beads, which is consistent with the study of Avci-Adali et al. [21].

Finally, the two ssDNA generation methods were evaluated qualitatively. First, the purification of the final products is essential to produce ssDNA free of impurities. Indeed, salts from buffers can alter the binding conditions of SELEX [21]. The maximum loss observed was 15% for ethanol precipitation and 17% for phenol-chloroform extraction, while a 30% loss of ssDNA is described in the literature [21]. Our improved results can be explained by the optimization of the protocol (washing step with 95% ethanol and the addition of a co-precipitant). It has been clearly demonstrated that streptavidin contamination has a negative impact on aptamer selection. Streptavidin can be responsible for cell aggregation, disrupting the SELEX process, especially when testing the binding of aptamer pools by flow cytometry [23]. Streptavidin can also constitute a good target for aptamers and may be responsible for the enrichment of non-specific and undesired sequences, leading to SELEX failure [15], a problem not encountered with lambda exonuclease. The effect of lambda exonuclease contamination on the selection process has not yet been demonstrated and should be further investigated. Within this method, no significant reduction in yield was observed between ethanol precipitation and phenol-chloroform extraction. Therefore, phenol-chloroform extraction should be strongly considered after ssDNA generation.

Moreover, the ssDNA generation method should not generate dsDNA contamination, as specific and highly affine ssDNA sequences may be lost [16]. According to the literature, it is not uncommon to observe contamination by dsDNA due to NaOH treatment after purification of ssDNA using streptavidin-coated beads, despite optimization [23,26], contrary to lambda exonuclease digestion. The generation of ssDNA by lambda exonuclease digestion is therefore advantageous both in terms of the yield of ssDNA generated and purity.

## 5. Conclusions

This work provides useful information in addition to previous studies [24,25], by comparing the two most commonly used ssDNA generation methods. The impact of ethanol precipitation or phenol-chloroform extraction on ssDNA yields were studied through robust data on optimized conditions. The yields were documented by three ssDNA quantification techniques, demonstrating the accuracy of the gel technique for small quantities of ssDNA measurement. Based on the in-depth comparison proposed in this study, lambda exonuclease digestion gave better results than the purification of ssDNA using streptavidin-coated magnetic beads in terms of quantity and quality of ssDNA. Lambda exonuclease digestion also gives additional advantages over streptavidin-coated beads, in terms of time (1 h process for lambda exonuclease digestion vs. 3 h for purification of ssDNA using streptavidin-coated beads), cost (83 euros for 100 uses for lambda exonuclease digestion vs. 1500 euros for 100 uses for purification of ssDNA using streptavidin-coated beads) and ease of use [29,39]. In addition, lambda exonuclease digestion has already demonstrated efficient selection of many aptamers against various targets, providing faster and more specific sequence enrichment than purification of ssDNA using streptavidin-coated beads [26,40,41,42]. All these advantages make the lambda exonuclease digestion method a promising and efficient alternative for ssDNA generation. Its application within a SELEX process could facilitate the selection of functional aptamers and consequently the development of their applications in fundamental research, as well as for diagnostic or therapeutic purposes. Our results could be extended to wider molecular and biotechnology applications requiring a ssDNA generation step, including pyrosequencing, single-stranded conformation polymorphism analysis, and DNA chips [43]. 

## Figures and Tables

**Figure 1 mps-05-00089-f001:**
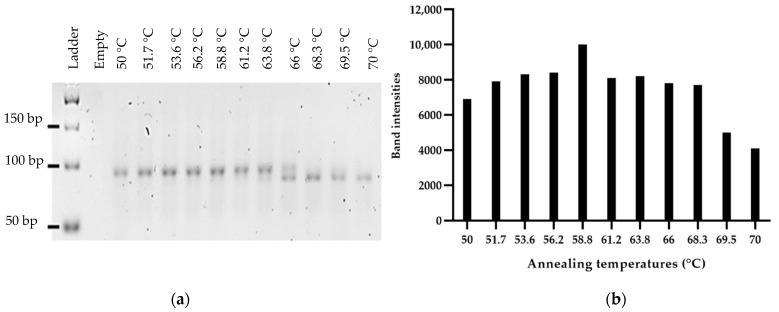
(**a**): Agarose gel electrophoresis (4% agarose, 90 V, 2 h in TAE 1× GelRed^®^) of the PCR products of the aptamer library with 20 cycles of amplification and annealing temperatures ranging from 50 °C to 70 °C. The legends above the gel correspond to the tested annealing temperatures. (**b**): Representation of the band intensities (ImageJ) as a function of the tested annealing temperature. Band intensities represent the amount of dsDNA produced.

**Figure 2 mps-05-00089-f002:**
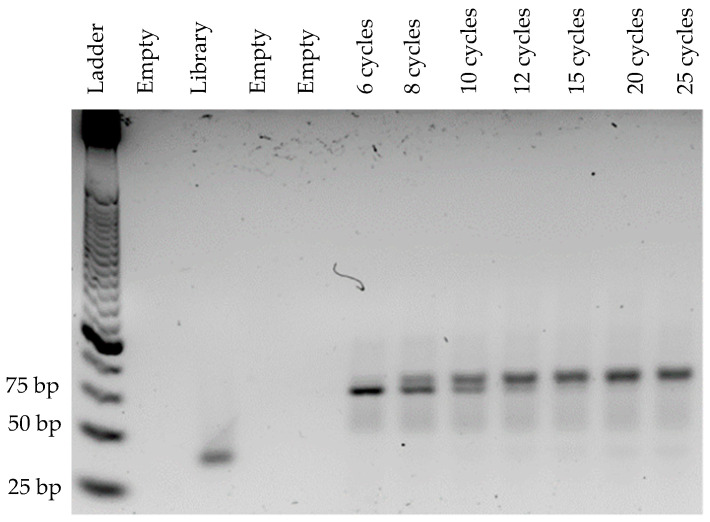
Agarose gel electrophoresis (4% agarose, 90 V, 2 h in TAE 1× GelRed^®^) of the PCR products, after variable numbers of amplification cycles (from 6 to 25 cycles) performed with an annealing temperature of 58 °C. The legends above the gel correspond to the number of amplification cycles.

**Figure 3 mps-05-00089-f003:**
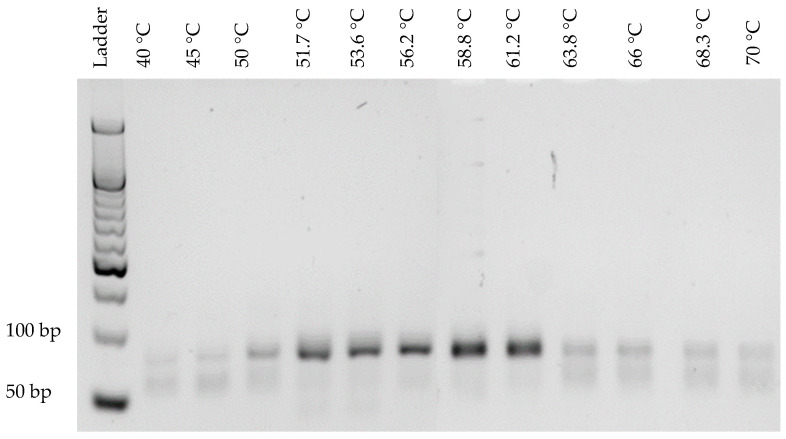
Agarose gel electrophoresis (4% agarose, 90 V, 2 h in TAE 1× GelRed^®^) of the PCR products of the aptamer library with 6 cycles of amplification and annealing temperatures ranging from 40 °C to 70 °C. The legends above the gel correspond to the tested annealing temperatures.

**Figure 4 mps-05-00089-f004:**
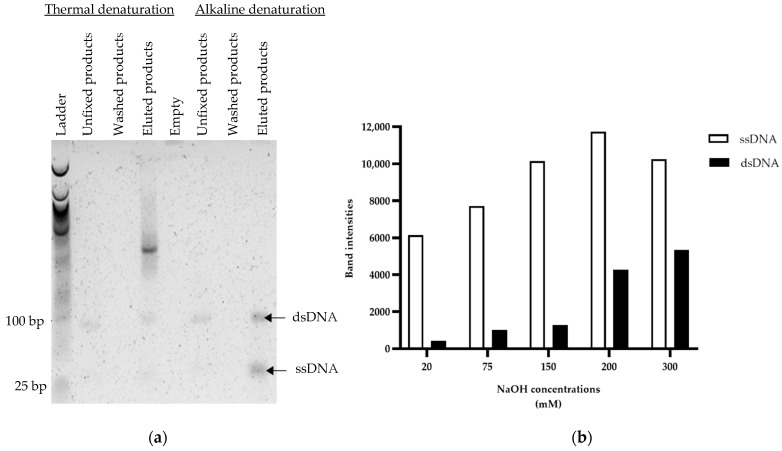
(**a**): Agarose gel electrophoresis (4% agarose, 90 V, 2 h in TAE 1× GelRed^®^) of eluted products following two elution methods (alkaline denaturation with 200 mM NaOH and thermal denaturation) with 20 mg of streptavidin-coated magnetic beads. (**b**): Representation of the band intensities (ImageJ), according to different NaOH concentrations tested (20, 75, 150, 200 and 300 mM NaOH) with 50 mg of beads. Band intensities represent the amount of ssDNA or dsDNA produced.

**Figure 5 mps-05-00089-f005:**
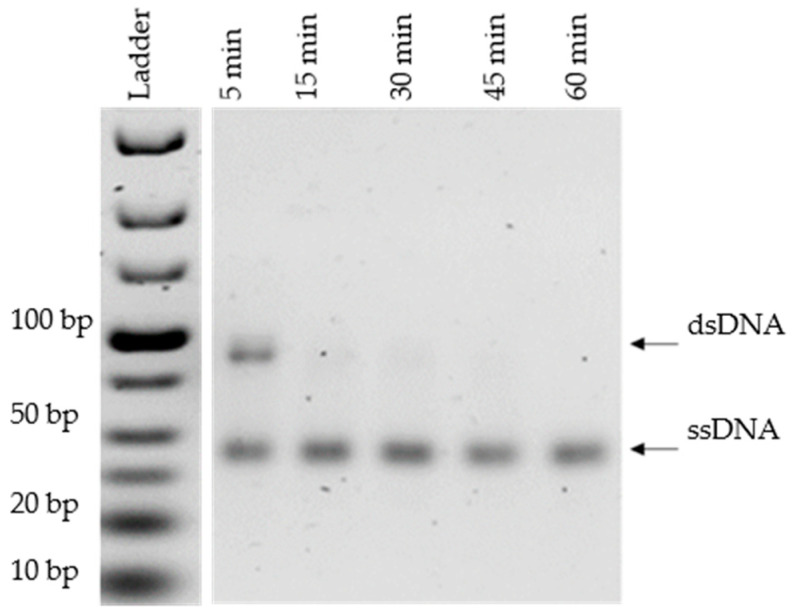
Agarose gel electrophoresis (4% agarose, 90 V, 2 h in TAE 1× GelRed^®^) of the final products after 5 to 60 min of incubation with lambda exonuclease. Products were revealed on a 4% agarose gel at 90 V for 2 h (TAE 1×, GelRed^®^).

**Figure 6 mps-05-00089-f006:**
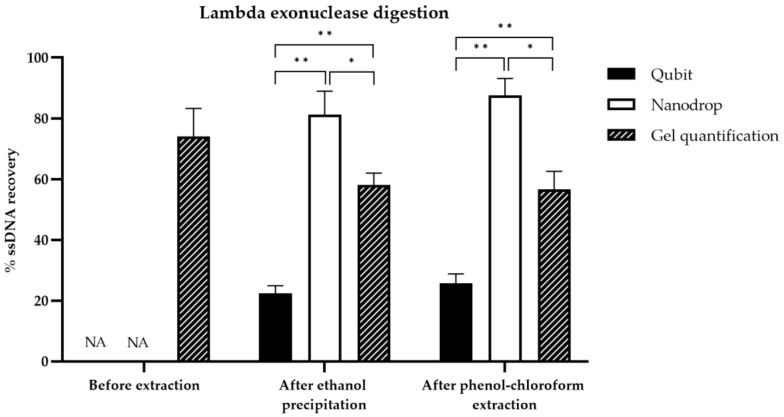
Histogram of ssDNA recovery quantified by Qubit, Nanodrop or gel quantification for lambda exonuclease digestion before and after two types of purification. The three ssDNA quantification techniques were compared in a peer comparison by Student’s statistical test. Asterisks represent significant differences according to Student’s *t* test (*: threshold of 0.01 or **: threshold of 0.001).

**Figure 7 mps-05-00089-f007:**
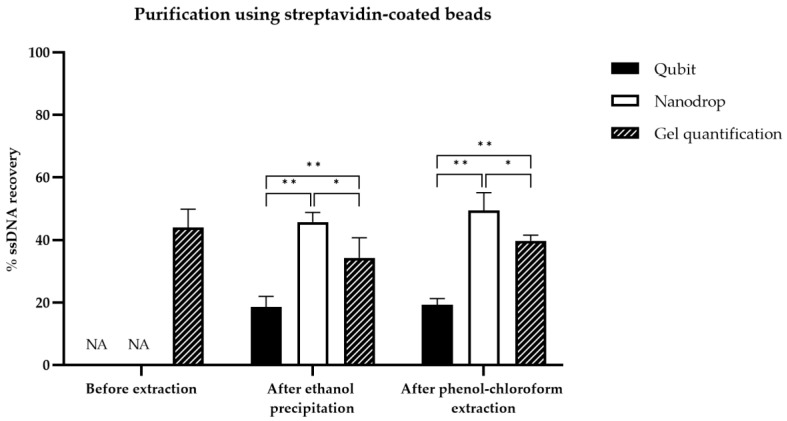
Histogram of ssDNA recovery quantified by Qubit, Nanodrop or gel quantification for purification of ssDNA using streptavidin-coated beads before and after two types of purification. The three ssDNA quantification techniques were compared in a peer comparison by Student’s statistical test. Asterisks represent significant differences according to Student’s *t* test (*: threshold of 0.01 or **: threshold of 0.001).

**Figure 8 mps-05-00089-f008:**
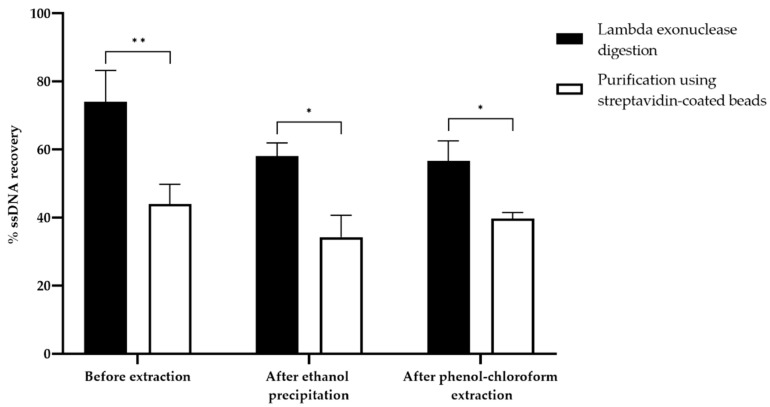
Histogram of ssDNA recovery quantified by gel electrophoresis for the two-ssDNA generation methods before and after two types of purification. Asterisks represent significant differences according to Student’s *t*-test (*: threshold of 0.01 or **: threshold of 0.001).

## Data Availability

Not applicable.

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
