# Peer review of "Optimized Lambda Exonuclease Digestion or Purification Using Streptavidin-Coated Beads: Which One Is Best for Successful DNA Aptamer Selection?"

_mps, 2022, doi:10.3390/mps5060089_

Round 1

Reviewer 1 Report

The authors presented a concise and practical manuscript to compare two ssDNA generating methods used in SELEX. The authors also compared different recovery methods and quantifications for ssDNA. Overall, the information provided in this manuscript is helpful for readers working on DNA SELEX.

Reviewer 2 Report

The manuscript “Optimized lambda exonuclease digestion or capture on streptavidin-coated beads: which one is best for successful DNA aptamer selection?" by Lisa Lucie Le Dortz and al. describes a comparison of two methods (streptavidin beads and lambda exonuclease) to generate single strand DNA (ssDNA) for aptamer selection. It also presents a comparison of three methods to quantify ssDNA. This comparison is not new, as indicated by the authors who fairly cite previous publications in the field. However, this comparison is, in my opinion, more complete and better described and therefore could be useful and worth publishing. Nevertheless, major revisions are required to correct errors in citations and especially to improve the presentation of results.

Major revisions:

1.     In the abstract and the introduction, the authors said: The generation of single-stranded DNA (ssDNA) is the most critical step of SELEX.” I do not agree, it is certainly not the most critical step (many scientists succeed to do it). Accordingly, it should be written as a critical step but not the most critical.

2.     In the introduction the first sentence is: “Studies conducted on HIV changed perspectives on the function of nucleic acids by demonstrating their ability to act as ligands through their three-dimensional structures, later termed aptamers by Elligton and Szotack in 1990”. I don't know which HIV studies the authors are referring to, but none of the original SELEX articles (like reference 2) are about HIV. Accordingly, I suggest that this sentence be modified and that reference 1, which serves no purpose here, be replaced with the reference describing SELEX published by Tuerk and Gold in 1990.

3.     In the introduction, authors must correct false or misleading information. Contrary to what is mentioned, asymmetric PCR don’t require ssDNA purification from polyacrylamide or agarose gel. Strand separation by denaturing gel does not generate both ssDNA and dsDNA but is the most efficient method to purify ssDNA. This method is time consuming but certainly not more expensive than the others if one takes into account the costs of purchasing the reagents (exonucleases or beads).

4.     The fact that gel-based quantification is more efficient is not clearly explained. For example, the authors said: "The use of a sample with a known ssDNA concentration (26 ng/μL) demonstrated the accuracy of the gel-based quantification method; therefore, this method was chosen for the comparison of yields." But this quantification of known sample is not provided as a figure.

5.     The authors should keep in mind that each conclusion is dependent on the experimental conditions that are used and should be written accordingly. It is therefore necessary to rewrite vague statements by being more precise. For example, when the authors said: “Clearly, in our case, the optimal number of amplification cycles corresponded to 6 cycles, which is consistent with the literature”. It should be replaced by: “Starting with a concentration of X µM of library, in our condition, the optimal number of amplification cycles corresponded to 6 cycles, which is consistent with the literature.” In the same way, when the authors said: “Second, a high amount of beads improved the capture of PCR products and the recovery of ssDNA downstream.” It should be replaced by: An increase of XX to YY mg of beads improved the capture of ZZ µg of PCR products and the downstream ssDNA recovery. “Through a time-course analysis, we propose an optimal digestion time of 30 min for lambda exonuclease, allowing the complete digestion of the phosphorylated strand.” It should be replaced by: Using a time analysis, we propose an optimal digestion time of 30 minutes for a complete digestion of xxµg/µl of phosphorylated strand by yy unit/µl of exonuclease lambda (because the time could be different if you use more enzyme or DNA).

Minor revision:

1.     Some correction of the English could improve the document. For example: “capture on streptavidin-coated beads” is not a method. It should be replaced by: purification of ssDNA using streptavidin-coated beads.

2.     Avoid obvious expressions. For example: “To generate ssDNA in sufficient amounts, enough phosphorylated or biotinylated sDNA must be available.”

3.     “The starting SELEX library consists of an infinite number of 350 possible sequences”. No, you can calculate that it is not infinite.

Reviewer 3 Report

The manuscript “Optimized lambda exonuclease digestion or capture on streptavidin-coated beads: which one is best for successful DNA aptamer selection?” by Le Dortz et al. presents some optimization of the library generation step of a SELEX. Overall, this manuscript feels like the authors want to publish their optimization for a given SELEX they are planning to perform. The comparison between lambda exonuclease and biotin-based purification was done before, so there is no novelty here.

As for the comparison of DNA concentration measurement methods, it is poorly presented (see below for some specific concerns and comments). It presents the gel-based quantification as the most accurate by stating “The use of a sample with a known ssDNA concentration (26 ng/μL) demonstrated the accuracy of the gel-based quantification method”, but nowhere in the paper do we see any actual numbers of ng/uL of measured DNA, we only see % recovery, so we cannot judge by ourselves. While gel-based measurements are probably the most sensitive (due to the use of DNA specific dyes), they are not necessarily the most accurate, especially when measuring higher concentrations. Also, the method used to measure the 2 ug input (coming from the PCR) is not mentioned. In the end, this aspect of the paper is not so relevant as the optimization of % recovery will not depend so much on the method used for quantification, which is more likely to depend on available materials and equipment in the lab performing the SELEX. That being said, it might be more appropriate to make a simple mention in the text with regards to the fact that gel-based quantification allows to measure small quantities of DNA, which can be useful when dealing with small amounts of DNA during the process of SELEX.

This brings me to the next big point, PCR seems to have been optimized only for the initial library preparation and then is presented with 6 cycles as optimal. However, following actual selection, only minute quantities of DNA will be left and 6 cycles of PCR will be insufficient to produce enough to allow detection afterwards. Some groups have used qPCR in that regard to follow the optimal number of cycles to be performed after each round (e.g. Navani et al., 2009, In Vitro Selection of Protein-Binding DNA Aptamers as Ligands for Biosensing Applications. Met Mol Biol). So, while this is a nice pre-optimization to do of the library preparation step (not considering other issues commented below), it is more or less relevant to the overall SELEX.

In short, while there are positive aspects to this manuscript that highlight the importance of fine-tuning SELEX conditions before starting it, overall the manuscript lacks novelty and has several other problems as mentioned above and below in my specific comments. For these reasons, I cannot recommend it for publication in “Methods and Protocols”.

Here are several additional comments, many minor points and several major points:

INTRODUCTION

Line 30:

Aptamers were first described simultaneously by two distinct research groups, not just Ellington & Szostak.

Cite Tuerk et Gold (1990) as well. DOI: 10.1126/science.2200121

Line 31:

I would replace “<100 nucleotides” by “generally <100 nucleotides”. There has been a couple of longer aptamers described in the literature before (<200 nt).

Line 33-34:

To avoid missing any interactions, I would change the wording for:

“...through various interactions like hydrogen bonds, electrostatic and Van der Waals (VdW) interactions”.

Line 44:

SELEX may potentially have more than just 3 steps. Notably, washing steps or negative selections. I suggest changing for the following “[…] selection process involving several steps, including: incubation of a ssDNA/RNA random library with a target, PCR amplification of the binding sequences, and generation of ssDNA/RNA […]”.

Line 47:

“Generation of ssDNA is the most critical step” this is highly debatable; I wouldn’t state it as fact. You cited McKeague’s lab at the end of your statement, but it isn’t what figures in the article cited, instead, in that paper we can read:

“... but it is unclear whether these inefficiencies impact the outcome of SELEX experiments.”

I suggest saying “is an important step” instead. Many would argue that the selection is the most crucial one, others could argue that PCR optimization is more crucial (library enrichment can be completely lost after only 5 cycles due to PCR artifacts (DOI: 10.1371/journal.pone.0114693).

Line 53:

“[…] leading to the loss of potentially binding sequences” I’m not sure this is a big deal. PCR produces thousands and thousands of copies for each candidate. Even if your ssDNA generation yield is barely 10%, in theory it would still be enough to not lose those candidates through more cycles, unless there is a sequence-based bias that would affect the % of molecules produced.

Line 55:

“is expensive because of modified primers” I doubt this is a big deal in the scale of a lab. Time, I think, is a more important factor than paying a little extra for a primer modification. Perhaps simply tune down with a wording like “...this method includes the use of more expensive primers that require chemical modifications.”

Line 59-60:

You mention “[...] is advised [...]” presumably this comes from a paper, add a reference.

MATERIALS AND METHODS

Line 79:

What does the “approximately 10^15 different ssDNA sequences” correspond to? The quantity used in assays presented in the manuscript? A typical synthesis for an oligo of that size would generally provide >1 nmole, which means >6 x 10^14 in total (corresponding, very roughly, to the provided number^of 10^15), but in the next paragraph authors mention that they use 2.5 uL of 1 uM as template (which means 2.5 pmoles, i.e. roughly 10^12). Authors should clarify this, for instance by indicating the number of sequences in parenthesis at the end of the sentence of line 94-95.

Line 97:

MgCl2 → MgCl2

Line 99:

“6 to 20 cycles”, meaning 6, 7, 8, … 19, 20 cycles were all tested? Or did you skip some of them?

Line 108:

“Final volume of 2 mL” I would rephrase to “20 PCR tubes each containing 100 uL (for a final volume of 2 mL)” (or whichever volume was used).

Line 119:

“with pre-washed beads” pre-washed with which buffer?

Line 127:

Remove the “of NaOH” in the parenthesis, we already know you’re referring to NaOH. Unnecessary redundancy.

Line 131:

sequence → strand

Line 138:

complete → highest. no enzymatic reaction can be 100% efficient even though it looks that way by eye.

Line 153, 154, and 157:

The “g” that stands for g-force should be in italic (g).

Line 180:

Please clarify what “amount of maximum ssDNA:1ug” means. Shouldn’t it be “total DNA before purification”?

RESULTS

Line 191:

after 66 → at temperatures higher than 66

Line 192:

(Figure 1) → (Figure 1a)

Lines 197-199:

Potential major flaw in data interpretation:

Authors write: “...Six amplification cycles were considered optimal based on the absence of non-specific amplification products...” However, the same band that is considered “non-specific product” here (the higher band) is likely the same one as that which was considered as the specific one on the gel from Fig 1 where the lower band is considered non-specific, since authors chose 58C as the optimal temperature (“...the lower band became predominant after 66°C...”).

It might be more obvious which band really corresponds to 80 bp if the 25 bp ladder was put right next to the PCR amplicons in Fig 2 (are the apparently empty wells really empty? They are not mentioned in the legend). Amplicons from the gel of Fig 1 a could also be helpful to help compare bands and determine if they are the same size; for instance by using the product of PCR at 58C (only higher band) and the product of the PCR at 69 (only lower band).

All mentions of amplicon sizes in the text should be revised accordingly, this does affect interpretation of results on several instances in the text.

Lines 238, 260 and Fig 4 and Fig3:

Even if authors clearly describe the ssDNA as “an intense band at 40 bp” on line 238, further use of the term “40 bp band” is confusing and should rather be indicated as ssDNA. Also, the size of dsDNA from the size marker should be indicated on the left side of the marker so that the readers can judge by themselves if the bands really correspond to the equivalent of a 40 bp (by the way, a DNA of 80 bases does not necessarily runs at the same position as a dsDNA of 40 bp; even if they are the same weight, because of the ss vs ds structures). Also, on Fig 3, the 80 bp and 40 bp (referring to the bands on the left) should not be beside the marker, but on the right side of the gel (close to the clearly visible bands) and would ideally be written as dsDNA and ssDNA (authors can keep 80 bp for the dsDNA if they want).

Line 262:

indicating the start of digestion → indicating partial digestion

Line 290:

the ssDNA → ssDNA

Line 292 (and 175):

“the use of a sample with a known ssDNA concentration” determined by which way? I think this information is important to mention in this sentence. If the information is provided by your oligonucleotide provider, visit their website to know which method they used.

Fig 5a: the graph appears to be “broken”, make sure the image comes out well at different zoom. More importantly, this figure poorly represents which technique is better. Authors state “Significant differences in post-purification yield between the three quantification techniques were confirmed by Student’s t-test (p<0.05) (Table S1, Figure 5)”, however, to compare the post-purification yield for each technique, the “pre-purification” DNA concentration should have been determined by each technique as well, and not only by gel purification, as is currently the case as presented in Fig 5. As I see it, the pairwise differences between Nanodrop and gel-based of Qubit and gel-based are coherent for ethanol precipitation and phenol-chloroform extraction, so I have to disagree with the statement.

Line 300:

were compared with each other → were compared

Line 309:

After purification only with ethanol, → After ethanol precipitation,

DISCUSSION

Line 350-351:

The starting SELEX library consists of an infinite number of possible sequences → A starting SELEX library consists of numerous possible sequences… (or, if you prefer, you can provide numbers, in the end, for most SELEX libraries, the number of different sequences is limited by the quantity of oligonucleotide used for the experiment)

Line 355-357

Please refer to which Figure you are talking about, in this case (Figure 1a)

Line 357

“the amplicon is likely due” mention what the amplicon is doing that you do not like (unexpected size).

Line 358

I don’t understand why a higher annealing would lead to more non-specific binding. It should be the other way around; the hotter it is, the harder it is for a primer to bind (or are you saying that higher temperature completely denatures the primer region rather than partially?). I don’t have the book you’re citing to verify, but I would ask you to go into more details or reword your sentence.

Line 368

gave the better yield → gave better yields

Line 381

I would add that lambda exonuclease is an enzyme that works with several buffers. The enzyme can be added directly inside a PCR mix after amplification without a need of a brand new mix. I know this is out of the scope of your paper, but it could be a useful information to add offhand since your paper is mostly about SELEX optimization. Maybe a sentence like “it could be worth to explore buffer compatibility with PCR as a time saver”.

Line 383:

Cite your figure at the end of your sentence.

Line 386-389

Maybe add 1 or 2 sentence about GelStain intercalation. Those kinds of stains bind dsDNA more efficiently than ssDNA (as reported by providers), but your article seems to point toward the stain being precise in both cases. Could you comment on this? It would be a useful information to your readers.

Line 407

Refer to your Figure.

Line 424-425

Syntax is deficient in this sentence. Perhaps it could be split it in 2, like this:

“… such as salts from buffers. These can alter …”

Line 431

You have more than one space between “...selection” and “as it...”

Also, this sentence is too long and is hard to follow. Split it in 2.

Line 434-435

leading to the failure of SELEX → leading to SELEX failure

Lines 435-437:

Contrary to streptavidin, lambda is not immobilized on beads, there is thus no reason for it to be the target of selected aptamers since putative lambda-nuclease aptamers that bind it would not be selected during the selection step. Simply delete this sentence or change for something like “...SELEX [15], a problem not encountered with lambda exonuclease.”

Line 440-441

“to avoid strep/lambda contamination” is that really a big deal? I’d add a sentence in how this could cause issues to the following SELEX cycles.

CONCLUSION

Line 449

provides essential information → provides useful information

Line 449-452

This sentence is too long. Split in 2 or 3.

Supplementary material:

What does “1/2; 1/3; 2/3” mean in the “Before purification” columns?

Reviewer 4 Report

Hello,

This article can help aptamer researchers.

I have a question:

Are you sure that the time of the electrophoresis in your research has written wright? Because, 2 hours at 90 V for agarose gel electrophoresis of small fragments of nucleotide seems very high.

Also, you'd better add more references in some parts of the discussion. For example, lines 485-487. 

Thanks 

Author Response

Answer to Reviewer 4

“Hello,

This article can help aptamer researchers.

I have a question:

Are you sure that the time of the electrophoresis in your research has written wright? Because, 2 hours at 90 V for agarose gel electrophoresis of small fragments of nucleotide seems very high.

Thank you for your positive feedback. The migration was performed at 90V for 2 hours in a 4% agarose gel to better visualize products between 10-150bp. This concentration was selected after testing different agarose concentrations (2%, 3% and 4%) for 2h at 90V. A 4% agarose gel was found to be optimal to visualize small single-stranded DNA.

Also, you'd better add more references in some parts of the discussion. For example, lines 485-487. 

We added three references in which lambda exonuclease was use to generate aptamers against a protein, a virus or two bacteria (lines 485-487). We also added another reference line 485.

Thanks »

Round 2

Reviewer 2 Report

In my opinion, the authors responded to the comments and therefore improved their article so that it could be published.

Author Response

We thank reviewer 2 for his last comments and are happy that our answers met his/her expectations.

Reviewer 3 Report

Overall I remain unconvinced of the value of this manuscript. As the authors note themselves, there is no novelty in terms of new methods. As for "best method evaluation", which would have value even if no novel method was described, it has already been done (the authors cite two papers for that matter). Authors argue that evaluation of ssDNA combined with PCR has not been described before. That being said, this is more or less to be done before any rigorous preparation of a SELEX project and does not represent any novelty or useful info (especially given that the only presented optimization is from the starting library at 0.1 uM, which can be amplified with only 6 cycles of PCR, but DNA following selection would be at much lower concentration and would thus require more cycles).

 With regards to the supposedly specific and non-specific bands from the gels in Fig 1a and Fig2, unfortunately the bands are relatively far from the ladders and the actual size of these bands could be confounded over one another. I do not necessarily suggest that authors should redo the gel (as I am not convinced this manuscript should be published, except perhaps if it is included together with an actual SELEX for which all the current results would be used for initial fine-tuning before performing SELEX; in which case this would present an excellent, well detailed, presentation of the initial library preparation and condition fine tuning), but if authors want to have more convincing results, they should either present the bands from Fig1a and Fig2 next to each other (withput any empty wells in between), so that we can see how the specific and non-specific bands described on each gel are of different or similar size, or at least put them close to the ladder.

With regards to measuring the yield of DNA, given that after lambda nuclease digestion there were salts and other contaminants that could affect measures, it might have been more appropriate to measure the original (pre-digestion) DNA with each method.

Author Response

Overall I remain unconvinced of the value of this manuscript. As the authors note themselves, there is no novelty in terms of new methods. As for "best method evaluation", which would have value even if no novel method was described, it has already been done (the authors cite two papers for that matter). Authors argue that evaluation of ssDNA combined with PCR has not been described before. That being said, this is more or less to be done before any rigorous preparation of a SELEX project and does not represent any novelty or useful info (especially given that the only presented optimization is from the starting library at 0.1 uM, which can be amplified with only 6 cycles of PCR, but DNA following selection would be at much lower concentration and would thus require more cycles).

Indeed, as mentioned, our objective was not to develop an innovative method. However, we have implemented a rigorous comparison approach, starting with the same amount of double-stranded DNA and optimizing the ssDNA generation protocols to improve the yield of ssDNA obtained. Moreover, our approach provides new information regarding the impact of ethanol precipitation or phenol-chloroform purification on the final yield of ssDNA. This allows to overcome a real gap in knowledge. Furthermore, a comparison of the three single-stranded DNA quantification techniques had not been published before. The data we provide is crucial as the amount of ssDNA before starting a new SELEX cycle is of paramount importance for the success of aptamer selection. 

Regarding the number of PCR cycles, it is clear that 6 cycles would be probably not sufficient to amplify aptamers in a selection round. In a selection round, the number of PCR cycles should be increased to obtain a maximal quantity of DNA with a minimal amount of non-specific products. Regardless of the number of PCR cycles, our ssDNA generation protocol can be applied with an input of 2µg of double-stranded DNA.

 With regards to the supposedly specific and non-specific bands from the gels in Fig 1a and Fig2, unfortunately the bands are relatively far from the ladders and the actual size of these bands could be confounded over one another. I do not necessarily suggest that authors should redo the gel (as I am not convinced this manuscript should be published, except perhaps if it is included together with an actual SELEX for which all the current results would be used for initial fine-tuning before performing SELEX; in which case this would present an excellent, well detailed, presentation of the initial library preparation and condition fine tuning), but if authors want to have more convincing results, they should either present the bands from Fig1a and Fig2 next to each other (withput any empty wells in between), so that we can see how the specific and non-specific bands described on each gel are of different or similar size, or at least put them close to the ladder.

The size of the bands could be questioned if it was based only on a visual estimate relative to the ladder bands. However, an objective measurement was performed, using Image J software, which gave the size of the bands in Figure 1 (95 and 87 bp) and Figure 2 (95 bp and 80 bp). The Image J software advantageously replaces a reading based on the ladder which always presents the risk of a reading tainted by the subjectivity of the human eye. The additional experiment carried out based on your precious comment relating to PCR conditions allowed to confirm that the DNA bands obtained had the expected size, thank you again for this.

With regards to measuring the yield of DNA, given that after lambda nuclease digestion there were salts and other contaminants that could affect measures, it might have been more appropriate to measure the original (pre-digestion) DNA with each method.

We agree with reviewer 3 that contaminants can interfere with DNA concentration measurement. This is why, in a preliminary study, after the purification of PCR products, the quantity of DNA was determined by the three methods: Qubit, Nanodrop and gel quantification. The 3 techniques gave results that were not significantly different, confirming a successful purification. Based on these results, we decided to opt for gel quantification as it allows to quantify the expected DNA product only, even if non-specific products and/or residual primers remain present.